# Single-molecule sensing of peptides and nucleic acids by engineered aerolysin nanopores

Chan Cao [1,2]*, Nuria Cirauqui [1,3], Maria Jose Marcaida[1,2], Elena Buglakova[1,4], Alice Duperrex[1], Aleksandra Radenovic [5] & Matteo Dal Peraro[1,2]*

Nanopore sensing is a powerful single-molecule approach for the detection of biomolecules. Recent studies have demonstrated that aerolysin is a promising candidate to improve the accuracy of DNA sequencing and to develop novel single-molecule proteomic strategies. However, the structure–function relationship between the aerolysin nanopore and its molecular sensing properties remains insufficiently explored. Herein, a set of mutated pores were rationally designed and evaluated in silico by molecular simulations and in vitro by single-channel recording and molecular translocation experiments to study the pore structural variation, ion selectivity, ionic conductance and capabilities for sensing several biomolecules. Our results show that the ion selectivity and sensing ability of aerolysin are mostly controlled by electrostatics and the narrow diameter of the double β-barrel cap. By engineering single-site mutants, a more accurate molecular detection of nucleic acids and peptides has been achieved. These findings open avenues for developing aerolysin nanopores into powerful sensing devices.

[1] Institute of Bioengineering, School of Life Sciences, Ecole Polytechnique Fédérale de Lausanne (EPFL), 1015 Lausanne, Switzerland. [2] Swiss Institute of Bioinformatics (SIB), 1015 Lausanne, Switzerland. [3] Department of Pharmaceutical Biotechnology, Universidade Federal do Rio de Janeiro, 21941-902 Rio de Janeiro, Brazil. [4] Skolkovo Institute of Science and Technology, Moscow 121205, Russia. [5] Institute of Bioengineering, School of Engineering, Ecole Polytechnique Fédérale de Lausanne (EPFL), 1015 Lausanne, Switzerland. *email: chan.cao@epfl.ch; matteo.dalperaro@epfl.ch

The use of artificial or biological pores with nanoscale dimensions is a successful strategy for the detection of molecules of similar size, like DNA, RNA, peptides, proteins, and polysaccharides[1–3]. The great success of their application in long DNA strand sequencing has ignited a widespread research interest in the fields of biology, physics, chemistry, and nanoscience[4]. More recently, nanopore technology has shown a great potential for single-molecule proteomics applications[5,6]. For instance, ionic current measurements through electrolyte-filled nanopores have been used to determine the shape, volume, charge, rotational diffusion coefficient, and dipole moment of individual proteins[7]. Therefore, to meet the demand of sensing various molecules, several materials have been explored, including transmembrane proteins[8], solid-state pores[9,10], and DNA/peptide origami[11,12]. Biological nanopores have spearheaded the approach, as they are superior in sensitivity and can be precisely manipulated by chemical modifications or genetic engineering.

Aerolysin from *Aeromonas hydrophila* is produced by bacteria as a β-pore-forming toxin (β-PFT). After years of investigation, aerolysin is currently one of the best characterized among all β-PFTs, as we have a deep molecular understanding of the structure of this pore when embedded at the host membrane. While aerolysin is secreted in an inactive form, it is activated upon proteolytic cleavage of the C-terminal peptide, being recruited to the host membrane by GPI-anchored proteins[13]. At this point, aerolysin oligomerizes into a mature heptameric pore that features a novel and unique fold, constituted by two concentric β-barrels formed by 2 inner β-strands and 3 outer β-strands per protomer, held together mainly by hydrophobic interactions[14]. This fold developed during the pore formation process is likely responsible for the ultra-stability of the aerolysin pore. Importantly, aerolysin is the prototypical member of a large β-PFT family whose members widespread among both eukaryotic and prokaryotic organisms and share a similar structure and assembly mode[15].

In addition to its microbiological relevance, aerolysin has been discovered to be a promising nanopore candidate, exhibiting a high sensitivity for the detection of DNA[16], peptides[17], and polymers[18], providing excellent current separation and a dwell time range suitable for accurate signal processing. Benefitting from its very narrow channel, aerolysin is able to directly discriminate single nucleobases[19], single amino acids[20], and methylated cytosines in the serum sample[21] in its wild-type (wt) conformation. Therefore, engineering its structure holds the potential to improve the accuracy for DNA sequencing and to develop novel single-molecule proteomic strategies. However, while the early resolution of the α-hemolysin (α-HL) pore and other more recent pores have promoted their wide adoption in emerging nanopore sensing techniques, barely any studies have focused on the structure-function relationship of the aerolysin capability for molecular sensing. Here, a series of mutants have been rationally designed and studied using molecular modeling and simulation based on recent aerolysin structures and models[14]. Promising mutants have been then expressed, purified, and reconstituted in lipid bilayer membranes for single-channel recording and molecular translocation experiments. By full integration of computational and experimental data, we could understand how the ionic conductance, ion selectivity, and translocation properties of the aerolysin pore are controlled at the molecular level. We have revealed that the current–voltage relationship of the engineered pores is determined by the electrostatics at the cap region where the double β-barrel is located. Interestingly, the dwell time of molecular translocation is strictly correlated to the diameter of the narrowest constriction of the aerolysin pore, demonstrating the importance of steric hindrance for molecular translocation. Altogether, the sensing determinants

of aerolysin are mainly controlled by residues at the distinctive double β-barrel cap, thus that any modification of the pore needs to preserve this unique sensing structure.

## Results

**Structural characterization of aerolysin pore mutants.** According to our previous study[22], R282 and R220 at the cap region (pore entry), and K238 and K242 at the stem region (pore exit) are the two main sensing regions of the aerolysin pore, defined by distinctive steric (i.e., the main two constriction points located around R220 and K238) and electrostatic features (i.e., various layers of positively and negatively charged residues are present in these regions, see Fig. 1a and Supplementary Fig. 1)[22]. Steric hindrance and electrostatics are the two most important factors for molecular sensing in nanopore single-molecule approaches[23]. Compared to α-HL that has only four charged amino acids, the lumen of aerolysin is highly charged (R282, D216, R220, D222, E237, K238, E258, K242, E254, K244, E252, and K246, see Supplementary Fig. 1). Therefore, to better understand the properties of these sensing regions, and how they modify the ability to sense different molecules, we rationally designed several single-point mutants, and used molecular dynamics (MD) simulations to study their structural variation, ion selectivity, and ionic conductance. In particular, to study the influence of the pore diameter, we replaced the amino acids at the sensing regions (R282, R220, K238, and K242) to alanine, as to enlarge the pore size (i.e., R282A, R220A, K238A, and K242A), or to tryptophan, for reducing the pore diameter (R282W, R220W, K238W, and K242W). Furthermore, to study the effect of electrostatics, the positively charged amino acid R220 was replaced to another positive (R220K), a negative (R220E), or a neutral (R220Q) amino acid with comparable side chain volume. Finally, analogous mutations were studied at the pore exit, i.e. K238R, K238N, K238E, and K238Q, as illustrated in Fig. 1b.

Model systems for MD were built following a procedure described in our previous work[22]. Briefly, the different pore mutants were prepared starting from an already equilibrated system comprising the aerolysin pore model embedded in a lipid bilayer, where single-point amino acid substitutions were introduced. After solvation in a 1 M KCl box and subsequent equilibration, all the systems were simulated for at least 200 ns of unrestrained MD applying a biased voltage of 150 mV. The pore model used in MD is a truncated version of the full-length pore in which only the membrane spanning β-barrel is present without the extracellular membrane binding domains (Fig. 1a). Previous and current results showed that this minimal model is able to recapitulated the physico-chemical properties of the full-length pore, hinting at the β-barrel structure as the key structure determining translocation properties in aerolysin. We analyzed the diameter along the pore lumen for the various engineered proteins during MD simulations (Fig. 1c, Supplementary Fig. 2 and Supplementary Table 1). As expected, replacements by alanine resulted in a wider diameter at the mutant position, and, in general, along the pore lumen. However, mutations at the pore exit (K238/K242) to alanine seemed to also affect the diameter at the pore entry, slightly decreasing it. Contrary to what we expected, replacements to tryptophan resulted in wider pores. This can be understood when comparing the side chain conformations along simulations (Supplementary Fig. 3). While the charged side chains of arginine and lysine residues extend completely to the lumen, interacting with water molecules and ions, the tryptophan residues turn back to the pore wall due to their partly hydrophobic nature. Interestingly, we observed again a decrease in diameter at the pore entry for mutations to tryptophan in other positions along the pore. Overall, it seems

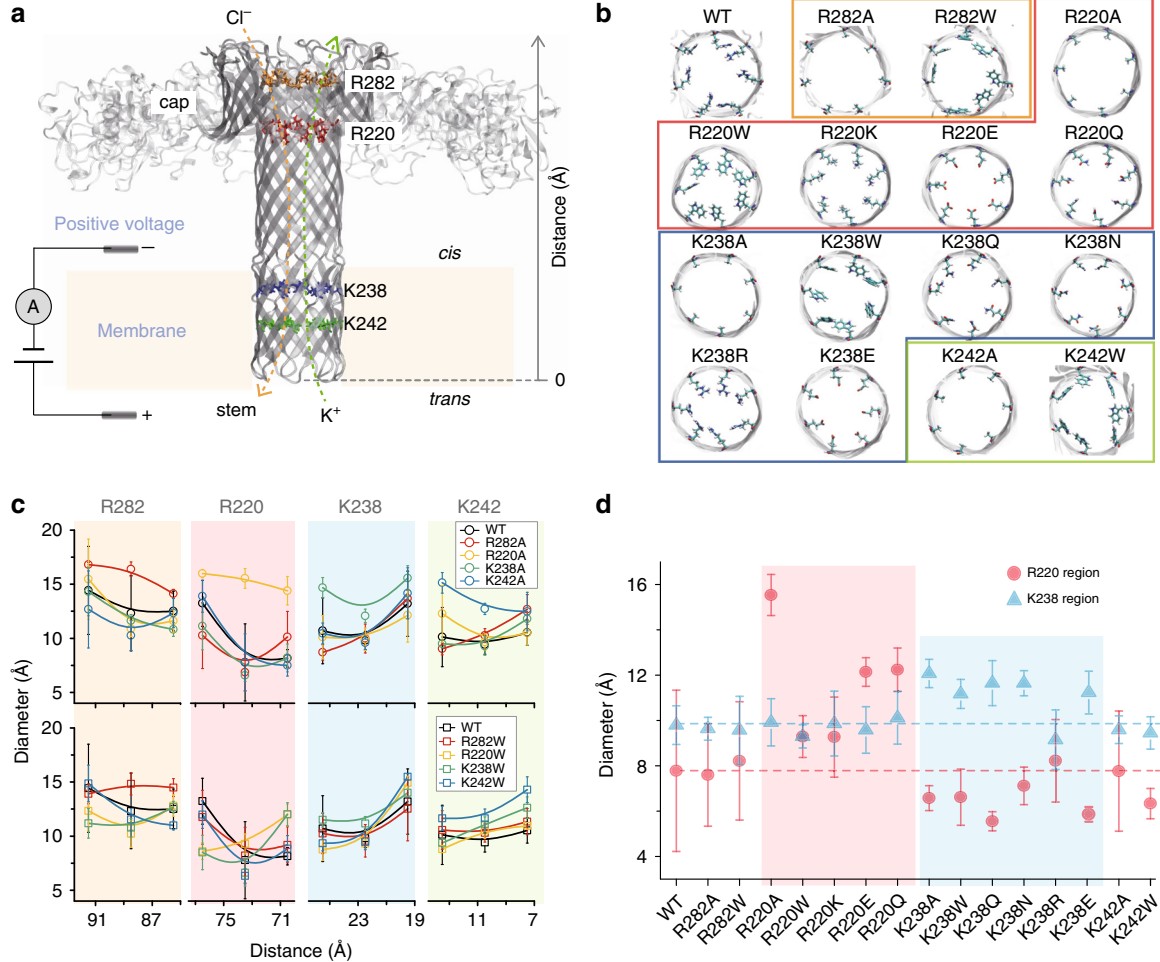

**Fig. 1** Structural characterization of aerolysin nanopore variants. **a** Structural model of aerolysin nanopore based on cryo-EM data[14,15]. The amino acids at the sensing regions are shown as sticks and colored as follow: R282 (orange), R220 (red), K238 (blue), and K242 (green)[40]. MD simulations were performed on a truncated pore model (i.e., residues 195–300 and 409–424, highlighted in solid dark gray). Here a typical simulation system is represented with positive applied voltage, i.e. a negative potential is applied at the cap side (*cis*) and a positive potential at the stem side (*trans*). The bottom of the pore is used as reference 0 Å distance in all the graphs. **b** Top view of the wt and mutants pore lumen. For wt, the view is focused at R220, and for the mutants, at the mutated region. **c** The diameter of wt and engineered aerolysin nanopores calculated with the PoreWalker server[35]. Distance ~88.5 Å represents the R282 position at the cap region, while a distance of ~73.5 Å indicates R220 also at the cap; ~22.5 Å corresponds to K238 at the stem, and ~10.5 Å is the K242 position also at the stem region. Each point represents the mean value ± st. dev. of at least 10 distinct structures extracted from MD simulations. **d** Calculated diameter of the various mutants at the two main sensing regions for 220 (red circles) and 238 (blue triangles) position

that an increase in pore diameter at the pore exit creates a slight decrease in diameter at the pore entry. When we analyzed the diameter at R220 and at K238 for all mutants (Fig. 1d), we observed that mutations at K238 are predicted to actually affect the diameter of the first constriction point (R220), but the contrary is not true. Moreover, we observed that R220 is the most rigid region in comparison to the rest of the pore, according to the calculation of root-mean-square fluctuations (RMSF) of the transmembrane inner barrel residues along the MD (Supplementary Fig. 4). Likely, as we previously discussed[14], this structural rigidity is linked to the higher stability of the double concentric β-barrel fold compared to the single β-barrel structure embedded in the lipid bilayer.

**Current conductance and ion selectivity of aerolysin mutants.** We calculated the open pore current of engineered pores at a fixed voltage (+150 mV), using a method already described by others[24] and in our previous work[22]. A positive voltage implies translocation of anions from *cis* to *trans* chamber, while cations

move from *trans* to *cis* (Fig. 1a). All current histograms are fitted with a Gaussian distribution (Supplementary Fig. 5) and fitted values are shown in Fig. 2a. Surprisingly, larger pores did not always display a higher open pore current. For example, R220A, with pore diameter two times wider than wt at the main constriction region, was predicted to have a reduction of 26% in open pore current, indicating that ionic conductance does not depends merely on the pore size. The electrostatics at the pore lumen also plays an important role, and a decrease of the positive charge at the pore entry (R220A) could be responsible for this reduced ionic current, likely by failing to efficiently capture negative chloride ions.

Based on this analysis, we selected relevant mutants for single-channel recording experiments: R282A, R220A, and R220W to check the importance of positive residues at the cap region; K238A, K238Q, K238N, and K238R to gauge the influence of mutations at the pore exit. The details of pore production and single-channel recording experiments are reported in the methods section[22] (see also Fig. 1a and Supplementary Figs. 6 and 7). The open pore current of the selected pores was measured

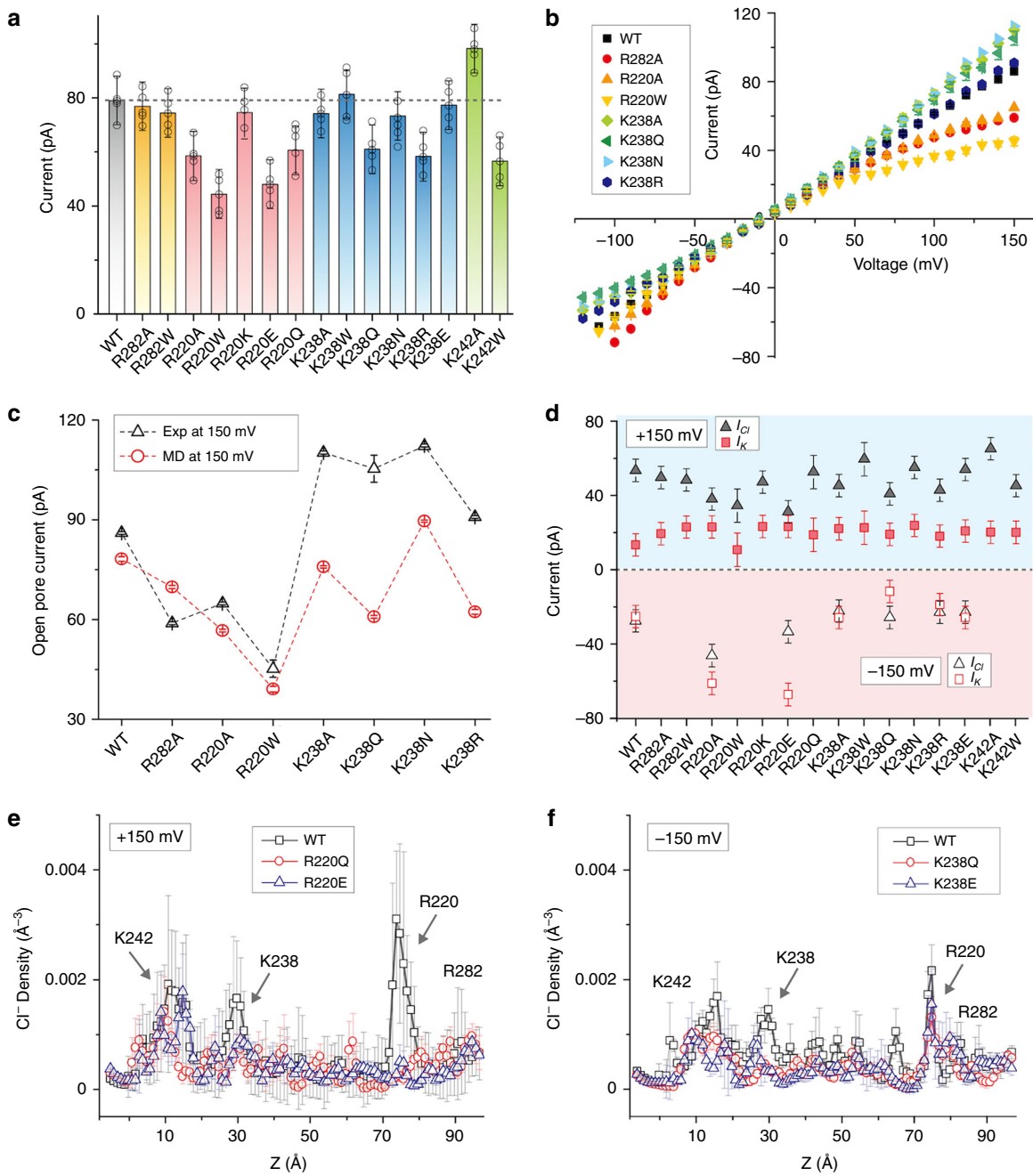

**Fig. 2** Open pore current and ion selectivity of aerolysin mutants. **a** Simulated open pore current at +150 mV in wt and engineered aerolysin pores. **b** Current–voltage relationships in aerolysin wt and mutants nanopores, where each point represents the mean value ± st. dev. of at least five independent experiments. **c** Comparison of open pore current at +150 mV calculated by MD simulations (red circles) with that obtained by single-channel experiments (black triangles). **d** The calculation of $I_{Cl}$ (black) and $I_K$ (red) in various engineered nanopores at 1.0 M solution of KCl electrolyte. At positive voltage, $I_{Cl}$ and $I_K$ are represented with solid triangles and squares, respectively, while shown with the open triangles and squares under negative voltages. **e** The averaged density of $Cl^-$ in wt (black squares), R220Q (red circles), and R220E (blue triangles) at +150 mV along the z axis of the pore, as defined by the local radii profile shown in Supplementary Fig. 8 during 200 ns MD simulation. **f** The density of $Cl^-$ in wt (black squares), K238Q (red circles), and K238E (blue triangles) at −150 mV

at different voltages (Fig. 2b), showing results similar to those obtained by MD simulations (Fig. 2c). The removal of a positive amino acid at the cap domain (R220A and R282A) resulted in a decrease in current at positive voltages, and an increase at negative ones. For example, the open pore current of R220A (48.4 ± 0.1 pA) decreased by 21% compared to the wt (61.5 ± 0.7 pA) at 100 mV, while the amplitude of current at −100 mV (−62.3 ± 0.1 pA) increased by 10% in comparison to the wt (−56.7 ± 0.1 pA). This trend fits very well the MD predictions,

therefore validating our models and the resulting molecular interpretations.

To further understand the nature of the current, we calculated the current of $K^+$ ($I_K$) and of $Cl^-$ ($I_{Cl}$) ions, separately. As observed in Fig. 2d (blue area), the open pore current at +150 mV is mainly produced by $Cl^-$ ions for all mutants. Therefore, in principle, the replacement of positively charged amino acids at the pore entry with neutral residues will reduce $I_{Cl}$. As expected, R220A showed a 28% decrease in $I_{Cl}$ while $I_K$ was

slightly increased compared to wt, which shows a good agreement with the 27% reduction in open pore current. The current calculations for wt, R220K, R220E, and R220Q further validated this hypothesis: the current of R220K (74.3 ± 9.0 pA) is in fact similar to that of wt (79.1 pA ± 9.0), while the current in R220Q decreased by 20% (63.4 ± 9.0 pA) and in R220E by 39% (48.0 ± 9.0 pA). On the other hand, mutations at the stem region showed no obvious modification in potassium translocation under the positive voltages, suggesting that the cap region is the controller of ionic current. To better understand this observation, MD simulations were also carried out at a negative voltage, where potassium ions are now captured at the pore entry. As observed in Fig. 2d (red area), contrary to the positive voltage results, none of the ion species does dominate the total current for all engineered pores and wt. However, the R220A and R220E showed a clear increase of $I_K$. This is explained by an increase in potassium uptake at the cap region, when compared to its stem uptake at positive potentials. Therefore, Fig. 2d highlights a behavior that has not been observed in other nanochannels, e.g. α-HL[24], that is, the total current is mostly controlled by the electrostatics at the cap region, which allows capturing either negative or positive ions depending on the applied voltage.

To better understand these differences in ion uptake and selectivity along the pore, we computed the averaged density of $Cl^-$ and $K^+$ in wt and mutants both under positive and negative voltages. The volume inside the channel was discretized along the pore axis (Supplementary Fig. 8), and the average was taken over all trajectories originated from the wt structure. As illustrated in Fig. 2e, the $Cl^-$ density in wt and R220Q/E mutants was calculated under a positive voltage (150 mV). The replacement of the positively charged amino acid in wt (arginine) to neutral (glutamine, R220Q) or negative (glutamic acid, R220E) residues dramatically reduced the density of $Cl^-$ at the mutated site. The reduction of $Cl^-$ density along the pore suggests, again, a reduction of $Cl^-$ uptake at the cap in the pore entry mutants. A similar effect was observed, as expected, for $Cl^-$ density in K238Q/E mutants at a negative voltage (Fig. 2f). Interestingly, we observed that the overall density of $Cl^-$ is reduced at this negative voltage when compared to the positive one, suggesting again the chloride capture is larger at the cap region than the stem. In addition, we could observe a strong increase in potassium selectivity for neutral mutants (R220Q and K238Q) and, for negative ones (R220E and K238E) (Supplementary Fig. 9). Furthermore, the influence of side chain size was studied by the same calculations for R220A and R220Q. As shown in Supplementary Fig. 10, no difference in $Cl^-$ density were observed, suggesting that the side chain volume did not significantly influence ion selectivity. Therefore, while the cap region is mainly responsible for ion uptake, ion selectivity in aerolysin is predicted to be mostly controlled by electrostatics, while pore size may have a minor role.

**Single-stranded DNA sensing is modulated by K238 mutations**. To test the sensing abilities of these engineered pores, we first performed translocation experiments for single-stranded DNA (ssDNA). The ssDNA used here is composed of 4 adenines, i.e., $dA_4$. As illustrated in Fig. 3a, $dA_4$ was added into the *cis* chamber after a single pore formed in the lipid bilayer, and then the translocation events were collected at various voltages ranging from +80 mV to +160 mV (raw current traces at +100 mV are reported in Fig. 3b). In comparison to wt, no signal was obtained for the R282A pores and very few events for R220A and R220W, revealing that the positively charged amino acids located at the cap region are crucial for capturing negatively charged molecules, such as ssDNA[25] (Fig. 3c).

The dwell time distribution for thousands of $dA_4$ events were shown in Fig. 3c and fitted by a falling exponential function. The fitted values are reported in Fig. 3d (red triangles). For mutations at the R220 position (R220A and R220W), the dwell time of events is really short (0.12 ± 0.02 ms and 0.15 ± 0.03 ms, respectively). Additionally, the dwell time of these two mutants remained unvaried under different applied voltages (Supplementary Fig. 11), demonstrating that the ssDNA did not translocate the R220A and R220W pores[26]. In order to better understand this observation, we calculated the electrostatic potential along the pore for wt and mutants under the same applied voltage of +100 mV, using the GCMC/BD Ion Simulator[27] (Fig. 3e). The R282A mutation reverses the potential at this point, resulting in a force working in the opposite direction and repelling the negatively charged DNA. This explains why there is barely any signal observed for the R282A variant. However, these data suggest that R282A mutants may work well for capturing positively charged molecules (see below). Additionally, we could also observe that mutations at the R220 region did not vary as much the potential at the pore entry, allowing DNA molecules to be captured but impairing efficient ssDNA crossing.

For mutations at the K238 position, the frequency of events did not change a lot except for K238Q but the dwell time of $dA_4$ was largely affected in all mutations (Fig. 3b, c and Supplementary Fig. 12). The frequency $f$ was calculated as $f = 1/\tau_{on}$, where $\tau_{on}$ is the inter-event interval for DNA crossing determined by single exponential fitting. Moreover, the dwell times experienced an exponential decrease at higher applied voltages, proving that $dA_4$ is actually translocated in these mutant pores (Supplementary Fig. 13). Notably, K238Q presents a unique property that allows $dA_4$ to translocate when the applied voltages are higher than +160 mV (Supplementary Fig. 13b). The three mutants, K238A, K238Q, and K238N, remove a positively charged amino acid at the pore exit, and therefore possess a similar variation in electrostatic profile when compared to wt (Fig. 3e). Therefore, the dwell time variation induced by mutations at the K238 region are not easily explained by electrostatic arguments, and could have a purely steric explanation[28]. When we compared the diameter size at the K238 region with the obtained dwell time of $dA_4$ translocation in different mutants (blue circles in Fig. 3d), no correlation was obtained. Then, considering that the chloride capturing at the cap side is the most important factor of ionic current, and that mutations at the pore exit modify diameter at the pore entry according to our previous results (Fig. 1d), we studied the diameter at the R220 region (black circles in Fig. 3d) and observed a correlation with dwell time. For example, K238Q constricted the diameter of R220 region to 5.6 ± 0.5 Å and therefore slowed the $dA_4$ translocation from 1.4 ± 2.1 ms to 1815.0 ± 127.5 ms (at +160 mV), which means the translocation speed of $dA_4$ through the K238Q pore is around 453 ms per base, more than 1000 times slower than wt pores. Furthermore, we observed an obvious reduction in the width of $I_{res}/I_0$ distributions ($I_{res}$ is the mean current blockade, while $I_0$ is the open pore current) for K238Q, demonstrating that having access to longer dwell times permits to evaluate the blockade current more accurately (Supplementary Table 2). Therefore, by making mutations at the stem region, it is possible to obtain significant modifications of the translocation properties at the cap region, allowing for a fine tuning of the molecular sensing properties.

**Peptide sensing by engineered aerolysin nanopores**. With the aim to extend the application of these pore mutants to a wider range of sensing tasks, we then measured their capabilities during

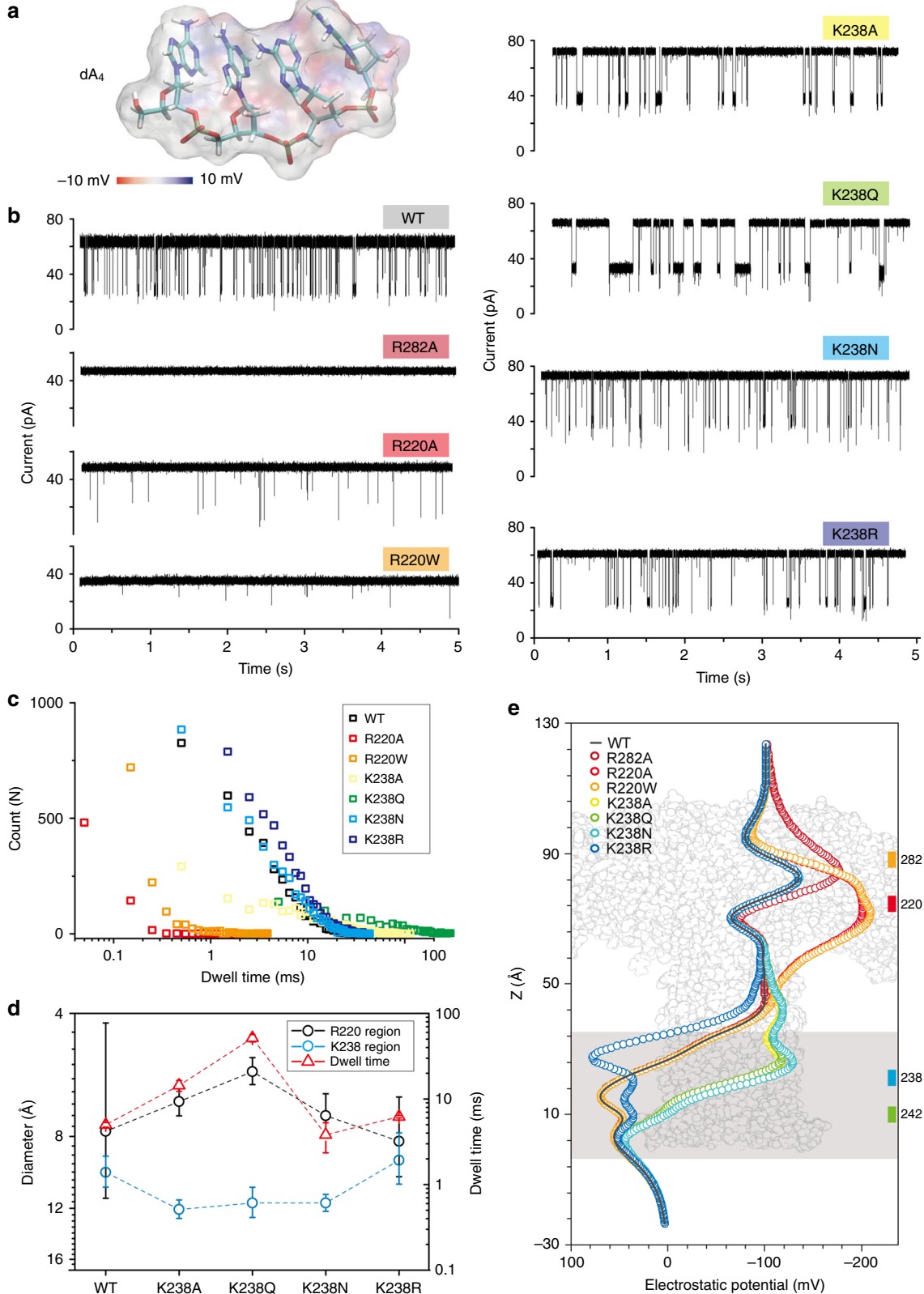

**Fig. 3** DNA sensing by various engineered pores. **a** Structure of dA$_4$ with its electrostatic potential mapped on the molecular surface. **b** Raw single-channel recording traces upon dA$_4$ addition into the *cis* chambers of wt, R282A, R220A, R220W, K238A, K238Q, K238N, and K238R mutants, respectively. The final concentration of dA$_4$ in the chamber was 2.0 μM. **c** Dwell time distribution of dA$_4$ events of the wt (black, 5000 events), R220A (red, 5000 events), R220W (orange, 3200 events), K238A (yellow, 3800 events), K238Q (green, 4200 events), K238N (sky blue, 5000 events), and K238R (blue, 5000 events) mutants under +100 mV voltage. **d** Comparison between dwell time of dA$_4$ (red triangles) and the pore diameter at the R220 (black circles) and K238 (blue circles) regions. **e** Electrostatic potential maps of wt, R282A, R220A, R220W, K238A, K238Q, K238N, and K238R along the pore lumen. The lipid membrane region is represented by a gray shadow, and the R282, R220, K238, and K242 regions are marked with the colors scheme used along the manuscript figures (see Fig. 1)

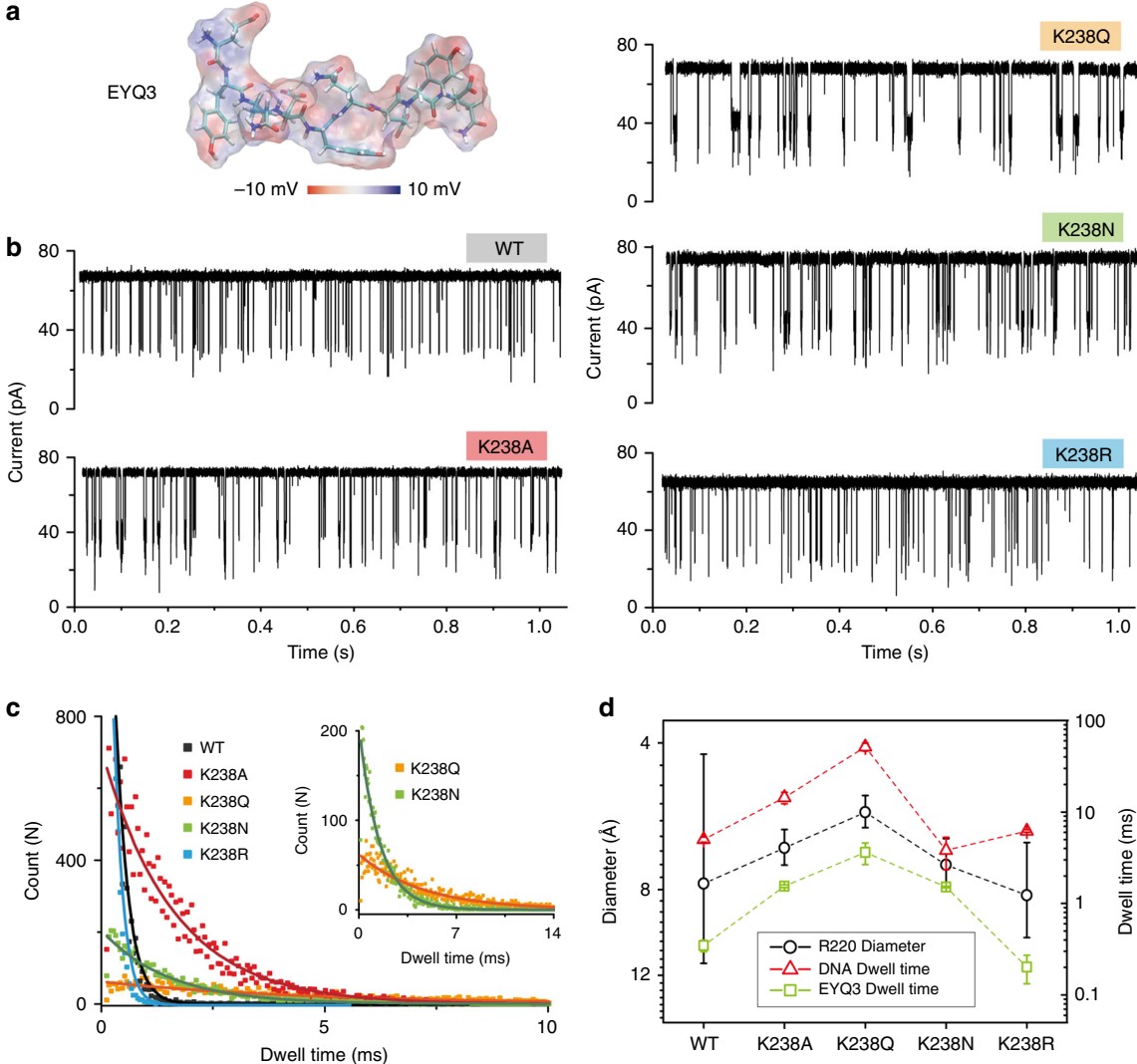

**Fig. 4** Negatively charged peptide sensing by aerolysin engineered pores. **a** 3D structure of EYQ3 with its electrostatic potential mapped on the peptide molecular surface. **b** Raw single-channel recording traces upon EYQ3 addition into the *cis* chamber of wt, K238A, K238Q, K238N, and K238R mutants, respectively. The final concentration of EYQ3 in the chamber was 2 μM. **c** Dwell time distribution of EYQ3 translocating through the wt (black), K238A (red), K238Q (orange), K238N (green), and K238R (blue) protein pores under +100 mV voltage. **d** Comparison between dwell time of EYQ3 (green squares), dwell time of DNA (red triangles), and the diameter at R220 region (black circles) for wt and pore mutants

peptides translocation. We first tested a negatively charged peptide, with sequence EYQEYQEYQ, named EYQ3 (Fig. 4a), which by design is expected to present similar translocation properties as ssDNA. Under a positive voltage (100 mV), we observed that K238A, K238Q, and K238N mutants showed significantly prolonged dwell time with respect to wt pores (Fig. 4b), as it was observed for $dA_4$ translocation. The fitted values are $1.6 \pm 0.1$ ms, $3.6 \pm 0.2$ ms, $1.5 \pm 0.1$ ms, and $0.4 \pm 0.1$ ms, respectively (Fig. 4c). K238R mutants on the other hand have a behavior ($0.2 \pm 0.2$ ms) similar to the wt pores. As for $dA_4$ experiments, the EYQ3 dwell time correlates well with the diameter at the R220 region (Fig. 4d), while the frequency of events remained constantly except for K238Q (Supplementary Fig. 14). Therefore, our results show how directed mutations at the K238 region can be used to modulate the molecular sensing properties of aerolysin nanopores for the negatively charged molecules in general, including both DNA and peptides. Notably, the dwell time of EYQ3 across the wt and mutant pores is almost 10 times faster than that of $dA_4$ (Fig. 4d). In this context, having discovered mutations (e.g.,

K238Q) that can significantly affect the translocation time by one order of magnitude has potential for developing aerolysin as a general single-molecule sensing device.

Next, we studied the translocation of polycationic peptides, using a short segment (47–57) of the HIV-1 Tat protein (Fig. 5a), with sequence YGRKKRRQRRR, which is highly basic and hydrophilic. Considering its features, we added HIV-1 Tat (47–57) into the *cis* side of the chamber and applied a negative voltage (100 mV) to drive it into the nanopore. For the wt pores, barely any signal was obtained, suggesting that aerolysin has a poor capability to capture highly positive peptides, which can be rationalized by the positive electrostatic potential at the mouth of the pore (Supplementary Fig. 15). In a previous study of α-HL, $K131D_7$, a mutant with a negatively charged ring at the *trans* entrance was able to capture the polycationic nanocarriers[29]. Here, we show that, as the positively charged amino acids in the cap region were replaced by the neutral amino acids, they are able to capture polycationic peptides. As shown in Fig. 5b, R282A, R220A, and R220W mutants exhibited an excellent capture ability

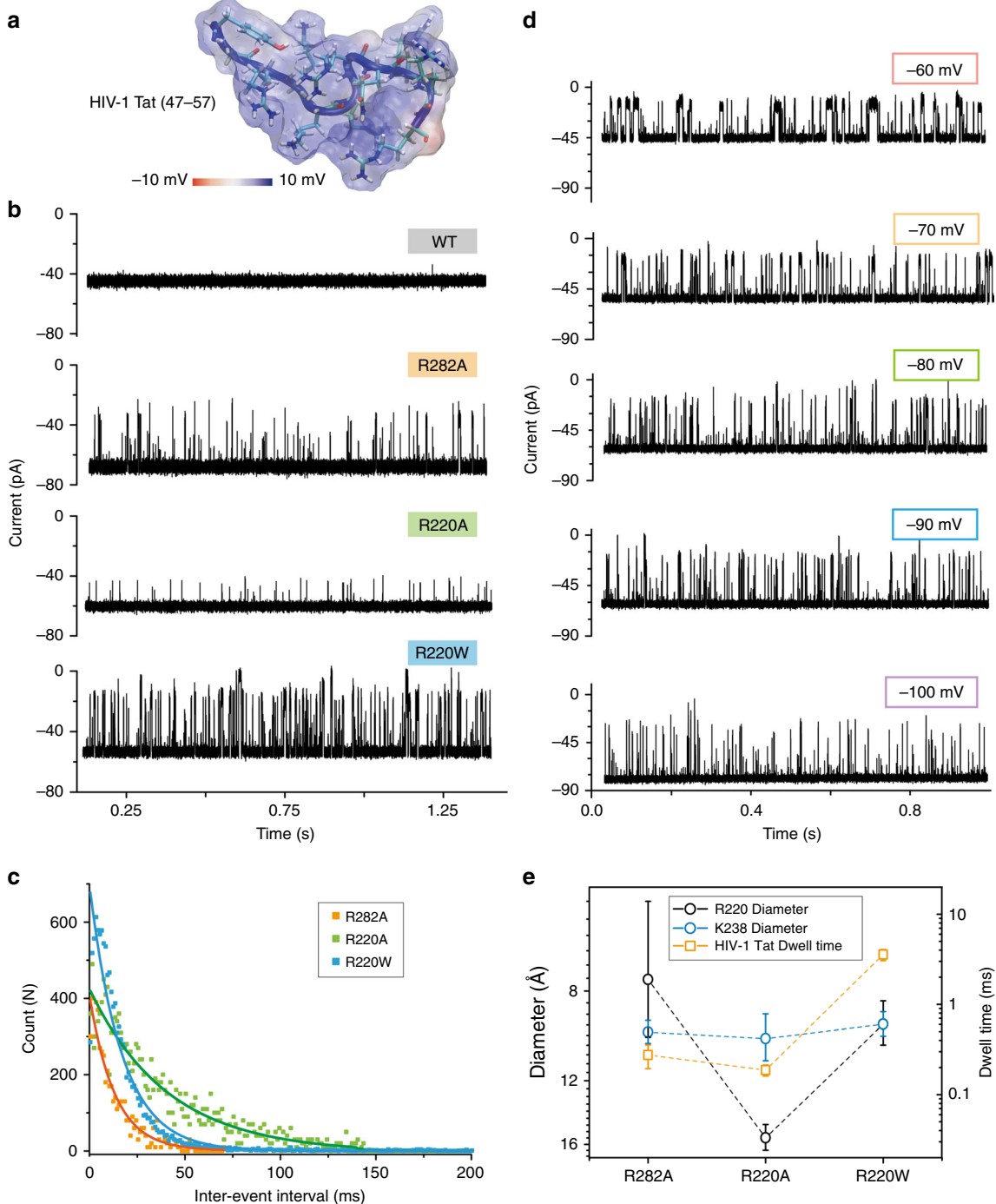

**Fig. 5** Positively charged peptide sensing by various engineering pores. **a** 3D structure of HIV-1 Tat (47–57) (PDB:1TAC) with its electrostatic potential mapped on the peptide molecular surface. **b** Raw single-channel recording traces upon HIV-1 Tat (47–57) addition into the *cis* side of the chamber for wt, R282A, R220A, or R220W, respectively. The final concentration of HIV-1 Tat (47–57) in the chamber was 2 μM. **c** Inter-event interval ($\tau_{on}$) distribution of HIV-1 Tat translocation on the R282A (orange), R220A (green), and R220W (blue) pore mutants. The values were determined by the single exponential fitting. Relative dwell times are reported in Supplementary Fig. 16. **d** Raw current traces of HIV-1 Tat (47–57) translocation through R220W at different voltages ranging (from −60 mV to −100 mV). **e** Comparison between dwell time of HIV-1 Tat (47–57) peptide (orange squares) and the diameter of R220 (black circles) and K238 (blue circles) region

for this polycationic peptide. The event frequencies of R282A, R220A, and R220W are $74.3 \pm 5.2 \, s^{-1}$, $24.5 \pm 1.8 \, s^{-1}$, and $59.4 \pm 0.5 \, s^{-1}$, respectively (Fig. 5c, dwell times for HIV-1 Tat (47–57) are shown in Supplementary Fig. 16). To confirm that the signal is caused by actual translocation events of HIV-1 Tat (47–57)

rather than transient bumping/interaction at the pore entry, we explored a range of applied voltages from −60 mV to −100 mV (Fig. 5d). It is visually clear how the dwell time decreases as the amplitude of applied voltages increases, demonstrating that the signals are caused by the translocations of peptides. When we

compared the dwell time of HIV-1 Tat (47–57) with the diameter of the R220 and K238 regions (Fig. 5e), it seems to fit the diameter at the first constriction point better. Therefore, these results show that engineered aerolysin nanopores can be also extended for sensing molecules that are prevalently positively charged, so as the HIV-1 Tat (47–57) peptide. Molecules with different chemical signature could in principle be specifically detected by using appropriate mutant pores. Such possibility makes the development of aerolysin-based nanopores very promising for single-molecule proteomic approaches.

## Discussion

Aerolysin differs from other biological nanopores mainly by its pore length, which is ~10 nm in comparison to the length of α-HL (~8.5 nm) and MspA (~8.0 nm). This longer length could be in principle a drawback for developing high sensitivity, and has likely led to a lack of attention for aerolysin as a molecular sensor. Based on our structural and functional knowledge of this pore, this unique feature seems more likely an advantage rather than a drawback. First, the longer pore together with the narrower diameter of aerolysin allows a slower translocation as to sense molecules without the need of an external support (e.g. additional DNA immobilization, adapter incorporation, or the processing enzyme), which is a common challenge to all nanopore sequencing approaches[30]. Second, this long pore may allow for broader and finer modifications in order to tune sensing of diverse molecules. Therefore, unraveling the structure-function relationship of aerolysin appears as a promising and powerful way to develop nanopores with desirable features. Herein, starting from our earlier observations[22], we rationally designed a large set of mutants studied by fully integrating in silico and in vitro experiments. Using this strategy, we managed to get insights into the translocation mechanism of aerolysin and were able to select a set of pore mutants highly promising for DNA and peptide sensing.

More important are however the molecular sensing properties which, according to our data, are mainly related to the diameter of the narrowest pore section[22]. We observed that the cap region is responsible for determining which molecule will be captured, acting therefore as a selectivity filter[25]. Thus, we were able to modify aerolysin properties as to natively capture under applied voltage not only negative, but also positive molecules, being R220W the most promising mutant for this task. On the other hand, the second constriction region (K238) was observed to be responsible for modulating molecule translocation time[31]. In this study, we extended this analysis to a broader set of mutants and, more importantly, we were able to determine the mechanism of this regulation. In spite of modifying the molecular translocation by modifications of size or electrostatics at the mutant region, mutations at the stem are able to influence the diameter at the cap, which is related to the differences in observed dwell time (Supplementary Fig. 17). Therefore, according to our data, the first constriction point is not only a selectivity filter, but also a sensitivity spot, and mutations at other regions along the pore (mainly K238) allow tuning its diameter for controlling the dwell time of different molecules. In particular, we found a mutant, K238Q, which, producing a smaller diameter at R220 and increasing the dwell time, could be used for sensing molecules which translocate faster than DNA, e.g. negative peptides.

In summary, the factors altering the open pore current, ion selectivity and structure variation of engineered aerolysin-based nanopores have been systematically studied and explained at the molecular level combining microscopic observations done in silico with macroscopic measurements in vitro. Altogether, our study proves that the sensitive diameter of aerolysin can be precisely controlled from 5 Å to 15 Å by site-direct mutagenesis, significantly extending the spectrum of detection capabilities for aerolysin nanopores. The ion selectivity can be tuned by the replacement of charged amino acids both at the double β-barrel cap domain, and at the stem pore exit region, providing a basic control of selectivity for specific biomolecules (i.e., ssDNA, and negatively and positively charged peptides). Finally, the strong correlation observed between the diameter at the pore cap region and the dwell time of mutants establishes the unique double β-barrel domain of aerolysin pores as the most important structural motif for molecular sensing, opening more precise rational avenues for tuning aerolysin sensing capabilities.

## Methods

**Molecular modeling and simulations**. The aerolysin pore was modeled using a previously prepared equilibrated system comprising an aerolysin pore model embedded in a lipid bilayer[22], where single-point amino acid substitutions were introduced. The pore structure in the initial system comprised the membrane spanning β-barrel (i.e., the membrane binding domains were not considered in the simulations), which remained stable for all MD simulations like the complete pore unit. The membrane bilayer was modeled by 1-Palmitoyl−2-oleoylpho-sphatidylcholine (POPC) lipids using the CHARMM-gui server[32]. Further details about the preparation and equilibration of the starting system can be found in our previous paper[22]. The mutant pores embedded in the equilibrated membrane were afterwards solvated on an $11 \times 11 \times 15$ nm 1 M KCl water box. After minimization using the steepest descent algorithm (integration step of 1 fs and 1000 kJ mol$^{-1}$ nm$^{-1}$ of maximum force), the solvent box was equilibrated for 0.1 ns (integration step of 2 fs), using position restraints for the protein and lipids (i.e., 10 kcal mol$^{-1}$ Å$^{-2}$ for protein backbone heavy atoms, 5 kcal mol$^{-1}$ Å$^{-2}$ for protein side chain heavy atoms, and 2.5 kcal mol$^{-1}$ Å$^{-2}$ for restraining the lipid tail and lipid head groups close to the membrane surface). All MD simulations were run using the GROMACS software (version 4.8)[33], with the CHARMM36 force field[34], the SHAKE algorithm on all the bonds involving hydrogen atoms, and Particle-Mesh Ewald, treating the electrostatic interactions with periodic boundary conditions. We chose an integration step of 2 fs. A temperature of 22 °C was controlled with the Nose-Hoover thermostat and the Parrinello-Rahman method was used for semi-isotropic pressure coupling. Biased voltage were applied with the z-dimension kept fixed, and with no need for using further position restraints[22]. The backbone root-mean-square deviation (RMSD) along the simulations can be found in Supplementary Fig. 18, where we could observe as previously reported for wt[22], a fast equilibration of all mutants within the first 10 ns of simulation. All systems were simulated for at least 200 ns.

The diameter of the pore along the MD simulations was calculated using the PoreWalker software[35], taking into consideration only the last 150 ns of the trajectories and selecting a frame every 10 ns, using a window size of 3 Å for the calculation. The ion current was calculated as described in our previous manuscript[22], following a method already described by others[24]. It should be noticed that due to the known overestimation of KCl bulk conductivity by the CHARMM force field[24,36–38] that we adopted, our approach can overestimate the absolute current value by a factor of 10 to 40%. The electrostatic potential was calculated using the GCMC/BD Ion Simulator[27], with an applied 100 mV voltage and using an implicit membrane model of 4 nm thickness. As input, we used for each pore mutant the full-length conformation, while the relative truncated versions of the pore mutants were used in MD simulations. Although using slightly different models and conditions, both calculations provided a consistent description of the electrostatic potential inside the pore barrel. Once oriented with respect to the membrane plane using the PPM server[39], the pore coincides to the membrane position observed during MD simulations. The GCMC/BD Ion simulator considers the solvent and lipids implicitly. The electrostatic surface was calculated by using the PBEQ software, using the same protein conformations as for the GCMC/BD Ion simulator, and following the parameters described in our previous manuscript[22]. For ion density, calculations were performed on the last 150 ns, where all mutants reached equilibration. All trajectory frames were fitted to the starting wt pore structure. Calculation of density for water, Cl$^-$ and K$^+$ was performed with VMD plugin volmap density, using a resolution of 1 Å, a step of 100 ps, and averaged as to obtain mean and standard deviation. We used a method described in previous work[24] to calculate radius of the pore cavity occupied by water and average density of Cl$^-$ and K$^+$ inside the cavity. The location of the pore axis is determined by the center of mass of the protein, then the volume was sliced into layers of 1 Å and the radius was found by the iterative process: first, the radius was assigned to be 1 Å and then it was increased by 0.5 Å while the ratio of water/non-water atoms inside the added ring was larger than 25%.

**Expression and purification of recombinant aerolysin mutants**. The aerolysin full-length sequence was cloned in the pET22b vector with a C-terminal hexa-histidine tag to aid purification. The QuikChange II XL kit from Agilent Technologies was used for performing site-directed mutagenesis on the aerolysin gene, following manufacturer's instructions. Recombinant proteins were expressed and

purified from BL21 DE3 pLys *E. coli* cells. Cells were grown to an optical density of 0.6–0.7 in Luria-Bertani (LB) media. Protein expression was induced by the addition of 1 mM isopropyl β-D-1-thiogalactopyranoside (IPTG) and subsequent growth over night at 20 °C. Cell pellets were resuspended in lysis buffer (20 mM Sodium phosphate pH 7.4, 500 mM NaCl) mixed with cOmplete™ Protease Inhibitor Cocktail (Roche) and then lysed by sonication. The resulting suspensions were centrifuged (12.000 rpm for 35 min at 4 °C) and the supernatants were applied to an HisTrap HP column (GE Healthcare) previously equilibrated with lysis buffer. The proteins were eluted with a gradient over 40 column volumes of elution buffer (20 mM Sodium phosphate pH 7.4, 500 mM NaCl, 500 mM Imidazole), and buffer exchanged into final buffer (20 mM Tris, pH 7.4, 500 mM NaCl) using a HiPrep Desalting column (GE Healthcare). The purified proteins were flash frozen in liquid nitrogen and stored at −20 °C.

**Single-channel recording experiments.** Phospholipid of 1,2-Diphytanoyl-*sn*-glycero-3-phosphocholine powder (Avanti Polar Lipids, Inc., Alabaster, AL, USA) was dissolved in decane (Sigma-Aldrich Chemie GmbH, Buchs, Switzerland) for a final concentration of 1.0 mg per 50 µl. Purified protein was diluted to the concentration of 0.2 µg ml$^{-1}$ and then incubated with Trypsin-agarose (Sigma-Aldrich Chemie GmbH, Buchs, SG Switzerland) for 2 h at 4 °C. The solution was finally certificated to remove trypsin. Phospholipid membranes were formed across a Delrin bilayer cup (Warner Instruments, Hamden, CT, USA), which separated the chamber into two part, *cis* and *trans*. After added aerolysin protein into the *cis* chamber, it could self-assemble to form a heptameric pore in the membrane. The electrolyte used here is 1.0 M KCl solution buffered with 10 mM Tris and 1.0 mM EDTA, titrated to pH = 7.4. Two matched Ag/AgCl electrodes were used to record the ionic currents. Then, the current traces were amplified and measured with a patch clamp amplifier (Axon 200B) equipped with a Digidata 1440 A A/D converter (Molecular Devices, Sunnyvale, CA, USA). The signals were filtered at 5 kHz and acquired with Clampex 10.4 software (Molecular Devices, Sunnyvale, CA, USA) at a sampling rate of 100 kHz. The data were analysed using Clampfit and OriginLab 8.0 (OriginLab Corporation, Northampton, MA, USA) software.

## Data availability

All data supporting the findings of this study are available within this paper and its Supplementary Information file. Any additional data are available from the authors.

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

## Acknowledgements

This research was supported by the Swiss National Science Foundation and Novartis Foundation for Medical-Biological Research (to M.D.P.), the European Union's Horizon 2020 Research and innovation programme under the Marie Skłodowska-Curie grant agreement No. 665667 (to C.C.). A.R. acknowledges the support from Swiss National Science Foundation (Grant number, BSCGI0_157802). We thank Sunhwan Jo for discussion on electrostatic potential maps and Sylvia Ho and Gisou van der Goot for support with aerolysin mutants production.

## Author contributions

C.C. and M.D.P. conceived the idea and designed the study; C.C. designed and performed the single-channel recording experiments, collected, and analyzed the data; C.C. and N.C. designed, conducted, and analyzed molecular modeling and simulations; E.B. contributed to molecular simulations; A.R. supported single-channel recording

experiments; C.C., M.J.M. and A.D. produced pore mutants; C.C., N.C., M.J.M. and M.D.P. interpreted the data; C.C., N.C. and M.D.P. wrote the manuscript with input from all authors; M.D.P. managed and supervised the project.

## Competing interests

The authors declare no competing interests.
