## [Peer Review File · Nature Communications]

Reviewers' Comments:

Reviewer #1:

Remarks to the Author:

In this manuscript, Chan Cao and co-workers report a comprehensive study of the effect of point mutations of transport of charge species through biological nanopore aerolysin. Using the all-atom molecular dynamics approach, the authors first characterize the local shape and ion conductivity of sixteen aerolysin variants that differ from one another by single amino acid in each of the six protomers. The specific mutations were chosen to alter the charge and the size of the amino acids in the four narrowest parts of the aerolysin stem. The simulation results revealed a complex dependence of the local pore size on the properties of the amino acid substitutions, reflecting an interplay of electrostatic and steric interactions. A smaller set of mutants was chosen for experimental electrophysiology recordings that examined the translocation of short DNA oligomers and of negatively charged and positively charged peptides. The experimental outcome clearly shows that point mutations can dramatically affect both the rate at which the molecules are captured by aerolysin and the duration of time the molecules spend blocking the current. In the case of DNA, one particular mutation extended the dwell time by a factor of 500. Point mutations have also enabled detection of translocation of cationic peptides.

Overall, the study provides a treasure trove of data that will be invaluable in future efforts to engineer aerolysin for molecular sensing applications. The manuscript is clearly written and can be followed by a non-specialist. The manuscript, however, contains a number of inconsistencies that need to be addressed before it can be published.

Major:

1. There is some confusion about how the electrostatic potentials shown in Figure 3e and SI figures were computed. The authors state that they've used GCMC/BD server, but do not specify what was the input for the calculations. Did they use coordinates of the protein and the lipid bilayer from the trajectories and averaged the potentials somehow? Or was the calculation done on the crystal structure after introducing single amino acid mutations? In the latter case, the data might not be compatible with the MD simulations which used a truncated pore model and a proper lipid bilayer, which is expected to affect electrostatic potential distribution. How do the results of the GCMC/BD calculations compare with the electrostatic potentials extracted directly from the all-atom MD trajectories using the method described in Ref 26?
2. The dwell time data for dA4 shown in Figure 3d and 4d do not appear to match SI figure 12 for the K238Q mutant and the text. The dwell time in Fig 3d is below 100ms and above 1 s elsewhere.
3. Please add K_{on} data to Fig. 3, 4 and 5. Dwell time distributions can be moved to SI along with the K_{on} distributions.

Minor:

Page 4: Although the authors do use the same MD model as in their previous studies, it nevertheless needs to be briefly introduced when the simulations are first described (second paragraph on page 4). Specifically, it must be stressed that a truncated model of aerolysin nanopore was used for all simulations. Figure 1 should show the model actually used for simulations, along with the schematics of a full length nanopore.

Figure 1a and other figures in the manuscript: what motivated the authors' choice for the specific placement of the lipid bilayer with respect to the aerolysin cap?

On a related note, could the authors comment on the behavior of the lipid bilayer in the all-atom MD simulations? Does the bilayer drift up and down along the outer surface of the nanopore?

Caption to Figure 1e: specify the typical duration of the MD trajectories and explicitly state what system was simulated (truncated pore embedded in a lipid bilayer, not just 1M KCl)

Page 6, third line: the references to "inward" and "outward" direction implicitly assume a certain polarity of the transmembrane bias and ion flow. Please revise.

SI Figure 5: Please specify the sampling frequency of the ionic current data used to construct the histograms in the caption to the figure.

The two paragraphs spanning pages 6-8 are somewhat technical and repetitive, consider streamlining to focus on most interesting observations.

Caption to Figure 2a: "Simulated" is probably a better word than "Calculated".

The reference to the HIV-1 Tat peptide (Ref 31) is a bit unorthodox, as it does not describe transport through a biological nanopore and suggests that the peptide can go through a lipid bilayer by itself. Perhaps a better choice could be one of Liqun (Andrew) Gu papers, for example, ACS Nano 11: 1204

At the bottom of page 15, the authors say that the first constriction is also the sensitivity spot. Which data support this conclusion?

SI Figure 15: the figure axes and labels are not legible, please revise.

Reviewer #2:

Reviewer #2:

“Aerolysin nanopore sensing is controlled at the double beta-barrel cap” by Cao et al. is a thorough study of point mutation modifications to the aerolysin pore and its effects on DNA and peptide sensing. It is a well-written manuscript that covers both experimental and theoretical descriptions of the aerolysin pore. I was particularly impressed with the mutations that show the ability to detect the cationic peptide (HIV-1 Tat(47-57)) while the wild-type version of the pore could not sense this at all. This work opens the possibility of detecting a wide range of various peptides and given the recent work by Pigué et al. on the use of aerolysin for homopolymeric detection, I believe studying other more biologically relevant peptides will require mutated forms of aerolysin such as the ones discussed herein.

Specific comments I would like to see addressed:

1. The data in Figure 2C fits well with the MD simulations for the arginine replacements, but the fit is considerably worse for the lysine replacements (i.e. K238A, K238Q, etc.). Do you have any explanations for why this might be the case?
2. Figure 3C and elsewhere. Consider plotting lifetime distributions on semi-log axis. It is very difficult to see what is going on when most of the data occupies only ~10% of the graph window.
3. I would like to see the mean current blockade distributions for the peptides (EYQ3 and HIV-1 Tat(47-57)) and DNA. The data shown in Figures 4 and 5 (peptides) seem to suggest some multi-state structure. I would also be interested to see how tightly distributed the blockades are. Also for the DNA data shown in Figure 3, I would be curious to see how the increased dwell times in (i.e. K238Q) improve the mean current blockade distributions as compared to the wild-type data.
4. You might want to consider modifying the title to convey the fact that you performed single molecule peptide and DNA sensing with the mutated pore. This will make it more likely to be referenced in any future review articles on the subject.

Response to Reviewers

We thank the reviewers for their very positive assessment of our manuscript. Below, we respond point-by-point to their concerns. All the revised parts in the revised manuscript and supporting information are marked in red accordingly to our response.

Response to Reviewer #1

Major:

Q1. There is some confusion about how the electrostatic potentials shown in Figure 3e and SI figures were computed. The authors state that they've used GCMC/BD server, but do not specify what was the input for the calculations. Did they use coordinates of the protein and the lipid bilayer from the trajectories and averaged the potentials somehow? Or was the calculation done on the crystal structure after introducing single amino acid mutations? In the latter case, the data might not be compatible with the MD simulations which used a truncated pore model and a proper lipid bilayer, which is expected to affect electrostatic potential distribution. How do the results of the GCMC/BD calculations compare with the electrostatic potentials extracted directly from the all-atom MD trajectories using the method described in Ref 26?

A1. We thank the reviewer for having raised this point, which was not clear indeed in the original submission. As correctly guessed, the electrostatic calculations done using the GCMC/BD server and the MD trajectories are slightly different. While with the GCMC/BD server we used atomic coordinates for the entire pore (the solvent and membrane contributions are then implicitly taken into account), we used the truncated version of the pore for calculating the electrostatics from MD simulations.

We have included a brief explanation on the revised manuscript to make this point cleaner (page 19): "The electrostatic potential was calculated using the GCMC/BD Ion Simulator²⁹, with an applied 100 mV voltage and using an implicit membrane model of 4 nm thickness. As input, we used for each pore mutant the full-length conformation, while the relative truncated versions of the pore mutants were used in MD simulations. Although using slightly different models and conditions, both calculations provided a consistent description of the electrostatic potential inside the pore barrel. Once oriented with respect to the membrane plane using the PPM server⁴⁰, the pore coincides to the membrane position observed during MD simulations. The GCMC/BD Ion simulator considers the solvent and lipids implicitly."

It should be noted that it was not our intention to compare the results from the GCMC/BD server with those from MD simulations. We used MD simulations on the truncated version of the pores (for enhancing the accessible sampling) to initially characterize their structural variations, current conductance and ion selectivity in order to propose potential mutations to be tested experimentally. Using this explorative approach, which we acknowledge can have limitations as the models are not complete,

we observed that during ssDNA translocation, R282 and R220 positions (R282A, R220A and R220W) could not capture ssDNA efficiently. To further understand this observation, we then used atomic coordinates of the entire pore to calculate the electrostatic potential using the GCMC/BD server and better rationalize the findings. Thus, it was not exactly our aim to characterize in an absolute way the electrostatic features but rather to develop a quick and consistent protocol to identify relevant hotspots which could be targeted by protein engineering.

Nonetheless, we understand the concerns of the reviewer and thus we checked if the truncated pore model is significantly affecting the estimated electrostatic potential along the pore lumen. We calculated with the GCMC/BD server the potential of the wt pore using this time the truncated pore as used in MD simulations (**Fig. R1b**). When we compared these results with the MD results (**Fig. 1Rc**), we noticed that having an implicit or explicit model of the membrane, as well as the entire or truncated pore, is not affecting much the potential at the transmembrane region ($Z \sim 0 \text{ \AA}$). On the other side, the potential at the cap is affected by the removal of the extracellular domains to a larger extent, but in turn qualitatively conserves similar features, also considering the use here of different approaches and conditions. The potential is always very negative at the mouth of the pore but there are some variations at the double barrel region. Based on these approximate analysis, we can say that the truncated version of the pore is able to recapitulate the most relevant electrostatic features, but it is maybe not the most accurate model for more precise calculations. It has to be noted that MD simulations of the entire pore would be needed to estimate the actual potential of aerolysin, however given the large size and cost of these calculations, we decided not to embark on these for the specific exploratory role of MD in this particular study. We have reported a discussion of these new aspects in the text of the revised manuscript at page 4 and in the Methods section at page 19.

Figure R1. (a) Electrostatic potential for wt aerolysin, full length protein based on cryo-EM data, PDB: 5jtz, calculated using the GCMC/BD Ion simulator with applied voltage of 150 mV. (b) Electrostatic potential of the wt pore (in truncated form as used in MD simulations) calculated using the GCMC/BD Ion simulator with applied voltage of 150 mV. (c) Electrostatic potential of the wt pore in truncated form calculated from MD simulations with applied voltage of 150 mV. All frames of the trajectory were aligned to the first frame. Electrostatic potential was acquired using PMEpot VMD plugin with Ewald factor 0.25, then linear external potential was added with custom Python script. Electrostatic radius and profile were calculated according to the reference 26. The transmembrane (stem) and double barrel (cap) regions are indicated by a grey background.

--

Q2. *The dwell time data for dA4 shown in Figure 3d and 4d do not appear to match SI figure 12 for the K238Q mutant and the text. The dwell time in Fig 3d is below 100ms and above 1 s elsewhere.*

A2. We would like to better clarify this point that we think was misunderstood. The data shown in Figure 3d is the dwell time of dA₄ blockade events in wt and mutants at 100 mV while the Supplementary Figure 12 (Figure 13 in the new version of SI) includes dwell time of dA₄ for wt and mutants at different voltages (from 80 mV to 160 mV). If one compares the data in Figure 3d with the data at 100 mV in Supplementary Figure 12, they are exactly the same: 5.0 ± 0.3 ms, 14.4 ± 2.1 ms, 51.6 ± 5.0 ms, 3.8 ± 1.4 ms, and 6.3 ± 0.2 ms for wt, K238A, K238Q, K238N and K238R, respectively. In addition, we discussed the K238Q separately because it is different from all the others, as “(...) K238Q presents a unique property that allows dA₄ to translocate when the applied voltages are higher than 160 mV (Supplementary Figure 12b, Figure 13b in the new version of SI).” Similarly, the data showed in Figure 4d is the dwell time of EYQ3 translocating through wt and mutants at 100 mV.

--

Q3. *Please add K_{on} data to Fig. 3, 4 and 5. Dwell time distributions can be moved to SI along with the K_{on} distributions.*

A3. We now report the τ_{on} ($1/K_{on}$) distributions relative to Figures 3 and 4 here below and in the new Supplementary Figures 12 and 14. Since we did not discuss the capture ability in Figures 3 and 4, we think it is preferable not to include them into the main text. For Figure 5d, we followed instead the suggestion of the reviewer and replaced the dwell time distributions by τ_{on} distributions in the new Figure 5, and moved the dwell time distribution to Supplementary Figure 16.

Supplementary Figure 12 | Inter-event interval (τ_{on}) distribution of dA_4 translocating through the (a) wt, (b) K238A, (c) K238Q, (d) K238N, and (e) K238R pore mutants. The fitted values are 125.1 ± 8.2 ms, 117.8 ± 5.4 ms, 373.5 ± 42.6 ms, 83.7 ± 2.0 ms, and 145.9 ± 6.9 ms, respectively.

Supplementary Figure 14 | Inter-event interval (T_{on}) distribution of EYQ3 peptide translocating through the (a) wt, (b) K238A, (c) K238Q, (d) K238N, and (e) K238R pore mutants. The fitted values are 20.8 ± 1.2 ms, 20.5 ± 1.6 ms, 45.3 ± 5.4 ms, 22.6 ± 2.2 ms, and 26.8 ± 0.6 ms, respectively.

Minor:

Q1. Page 4: Although the authors do use the same MD model as in their previous studies, it nevertheless needs to be briefly introduced when the simulations are first described (second paragraph on page 4). Specifically, it must be stressed that a truncated model of aerolysin nanopore was used for all simulations. Figure 1 should show the model actually used for simulations, along with the schematics of a full length nanopore.

A1. We thank the reviewer for this observation, which we followed to ensure optimal reproducibility of our work. We have added in the revision a brief introduction of the simulation system and method in the main text at page 4: “**Briefly, the different pore mutants were prepared starting from an already equilibrated system comprising the aerolysin pore model embedded in a lipid bilayer, where single-point amino acid substitutions were introduced. After solvation in a 1 M KCl box and subsequent equilibration, all the systems were simulated for at least 200 ns of unrestrained MD applying a biased voltage of 150 mV. The pore model used in MD is a truncated version of the full length pore in which only the membrane spanning β -barrel is present without the extracellular membrane binding domains (Figure 1a). Previous and current results showed that this minimal model is able to recapitulated the physicochemical properties of the full-length pore, hinting at the β -barrel structure as the key structure determining translocation properties in aerolysin.**”

We also highlighted the model structure in Figure 1a with the truncated version of the pore we used in MD simulations, and updated the figure caption: “**Our MD simulations were performed on a truncated pore (i.e., residues 195-300 and 409-424, highlighted in dark gray).**” Finally, we realized that we forgot include the length of the simulations on the method section, which we have introduced now at page 18: “**All systems were simulated for at least 200 ns.**” As discussed above, we used the further analysis required by the reviewer to stress the fact that the truncated structure of the pore is a good minimal model able to capture the electrostatic features of the aerolysin pore.

--

Q2. Figure 1a and other figures in the manuscript: what motivated the authors' choice for the specific placement of the lipid bilayer with respect to the aerolysin cap?

A2. The placement of the pore with respect to the membrane is not dictated by any special motivation but by previous calculations and simulations performed in our lab. First, in previous works by us and our collaborators experiments were designed to map the exact transmembrane extension of the pore barrel. Second, the PPM server predicted this membrane position based on the physicochemical properties of the barrel. Finally, extensive MD simulations with different barrel positions were always consisting in proposing this final membrane-pore interface. More recently, the lipid density obtained in cryo-EM for wt aerolysin pore (*Nat Commun*, 2016, 7, 12062)

showed the membrane interface at the same region. Given this arrangement, we notice that no lipids are able to penetrate through the pore at any time during the MD simulations.

--

Q3. On a related note, could the authors comment on the behavior of the lipid bilayer in the all-atom MD simulations? Does the bilayer drift up and down along the outer surface of the nanopore?

A3. Based on the multi-microsecond trajectories accumulated so far in our lab, we never observed drifting of the membrane bilayer with respect to the barrel, hinting in turn to a very strong hydrophobic match that is anchoring the pore to the membrane.

--

Q4. Caption to Figure 1e: specify the typical duration of the MD trajectories and explicitly state what system was simulated (truncated pore embedded in a lipid bilayer, not just 1M KCl)

A4. We have included this information in Figure 1 and in the main text at pages 4-6.

--

Q5. Page 6, third line: the references to “inward” and “outward” direction implicitly assume a certain polarity of the transmembrane bias and ion flow. Please revise.

A5. Thanks for this correction, we revised them as follows (page 6): “A positive voltage implies **translocation of anions from cis to trans chamber, while cations move from trans to cis (Figure 1a).**”

--

Q6. SI Figure 5: Please specify the sampling frequency of the ionic current data used to construct the histograms in the caption to the figure.

A6. We specified the sampling frequency of the ionic current data used to construct the histograms in the caption of Supplementary Figure 5 as: “The histograms plotted here were taken over a 200 ns MD trajectory **with a sampling frequency of 10 ps. Then we used the adjacent-averaging (points of window is 500) to smooth the data by using the ORIGIN software as we did in a previous study².**”

--

Q7. The two paragraphs spanning pages 6-8 are somewhat technical and repetitive, consider streamlining to focus on most interesting observations.

A7. We have tried to reduce as much as possible the length of these paragraphs in the revised manuscript, however it has been difficult as we feel that all the observations therein reported are somehow interesting and would deserve to be reported and discussed.

--

Q8. *Caption to Figure 2a: "Simulated" is probably a better word than "Calculated".*

A8. We have corrected it.

--

Q9. *The reference to the HIV-1 Tat peptide (Ref 31) is a bit unorthodox, as it does not describe transport through a biological nanopore and suggests that the peptide can go through a lipid bilayer by itself. Perhaps a better choice could be one of Ligu (Andrew) Gu papers, for example, ACS Nano 11: 1204*

A9. Thanks, we deleted this reference and added the ACS Nano 11: 1204 in the revised text.

--

Q10. *At the bottom of page 15, the authors say that the first constriction is also the sensitivity spot. Which data support this conclusion?*

A10. This conclusion was supported by the observed correlation between the diameter at the R220 region and the dwell time of molecules translocating across the wt and mutant pores (Figure 3d, Figure 4d and Figure 5e). If we merge all these data (Fig. R2 below) it might be clearer to see. Previous works observed that mutations at the first constriction affected capturing, while mutations at the second affected duration (sensing). Here, we show that mutations at the second constriction decrease the diameter at the first constriction, in turn affecting sensing. Therefore, any modification along the lumen seems to eventually modulate the properties at the pore cap (first constriction), which is thus the key region of the aerolysin pore barrel.

Figure R2. The comparison of diameter and dwell time of DNA and peptides (negatively and positively charged) transport through wt aerolysin and various mutants.

--

Q11. SI Figure 15: the figure axes and labels are not legible, please revise.

A11. We enlarged the fonts in the revised Supplementary Figure 15 (Figure 17 in the new version of SI).

Response to Reviewer #2

Q1. The data in Figure 2C fits well with the MD simulations for the arginine replacements, but the fit is considerably worse for the lysine replacements (i.e. K238A, K238Q, etc.). Do you have any explanations for why this might be the case?

A1. We have thought about this, which is in fact the main discrepancy between *in silico* and *in vitro* data, however we do not have any definitive answer and we can only speculate on these differences. It is for sure not a force field problem as the K238R mutant has the same problem. Thus, it can be due to the peculiar structure of aerolysin as all the worse prediction affect the bottom of the barrel. Other biological nanopores that have been simulated to date present always shorter pore tracts. On the other hand, aerolysin has a very long pore barrel (~10 nm). Lysine residues are located in a region much more buried inside the barrel than arginines which are present at the mouth, thus that current calculations might be affected by poor ion diffusion and much longer simulations will be needed to fully capture these current features.

--

Q2. Figure 3C and elsewhere. Consider plotting lifetime distributions on semi-log axis. It is very difficult to see what is going on when most of the data occupies only ~10% of the graph window.

A2. We agreed with the suggestion of the reviewer and revised Figure 3c accordingly (see revised manuscript at page 10).

--

Q3. I would like to see the mean current blockade distributions for the peptides (EYQ3 and HIV-1 Tat(47-57)) and DNA. The data shown in Figures 4 and 5 (peptides) seem to suggest some multi-state structure. I would also be interested to see how tightly distributed the blockades are. Also for the DNA data shown in Figure 3, I would be curious to see how the increased dwell times in (i.e. K238Q) improve the mean current blockade distributions as compared to the wild-type data.

A3. We now report the additional data required by the reviewer here in the response and in the revised manuscript. Considering the open pore current of wt and various mutants are different, we plotted the I_{res}/I_0 distributions instead of mean current blockade distributions (I_{res} is the mean current blockade while I_0 is the open pore current) for peptides (EYQ3 and HIV-1 Tat(47-57)) and DNA. As shown in **Figure R3**, only one population was observed in wt and mutants for detection of EYQ3 peptide. However, the I_{res}/I_0 distributions of HIV-1 Tat(47-57) in R282A and R220W show multiple peaks, which means there are some multi-state structures present (**Figure R4**), as the reviewer expected. R220A only has one peak because the dwell time of HIV-1 Tat(47-57) is too fast to see any multi-state structure of the peptide. As for the

detection of dA_4 (**Figure R5**), it is obvious that the increased dwell times (i.e. K238Q) decrease the width of I_{res}/I_0 distributions as compared to the wild-type aerolysin. We also list the value extracted from the Gaussian fitting and the width of the half peak in a **Supplementary Table 2**. These data show that the longer dwell time results in a more accurate estimation of the blockade current.

Figure R3. The I_{res}/I_0 distribution of EYQ3 peptide crossing the (a) wt, (b) K238A, (c) K238Q, (d) K238N, and (e) K238R mutants.

Figure R4. The I_{res}/I_0 distribution of HIV-1 Tat(47-57) peptide translocating through the (a) R282A, (b) R220A, and (c) R220W mutants.

Figure R5. The I_{res}/I_0 distribution of dA₄ crossing the (a) wt, (b) K238A, (c) K238Q, (d) K238N, and (e) K238R mutants.

Supplementary Table 2. The I_{res}/I_0 and the width of half peak extracted by Gaussian fitting as EYQ3 peptide and dA₄ are translocating through the wt, K238A, K238Q, K238N and K238R pores, respectively.

	EYQ3		dA ₄	
	I_{res}/I_0	Width	I_{res}/I_0	Width
WT	0.500	0.040	0.450	0.005
K238A	0.550	0.033	0.520	0.009
K238Q	0.510	0.032	0.500	0.003
K238N	0.570	0.040	0.540	0.017
K238R	0.410	0.086	0.430	0.005

--

Q4. You might want to consider modifying the title to convey the fact that you performed single molecule peptide and DNA sensing with the mutated pore. This will make it more likely to be referenced in any future review articles on the subject.

A4. We thank very much the reviewer for this advice. Following her/his suggestion we have changed the title in “Single-molecule sensing of peptides and nucleic acids by engineered aerolysin nanopores”.

Reviewers' Comments:

Reviewer #1:

Remarks to the Author:

The authors have addressed all points raised in the previous round of review and the manuscript can be accepted, in principle, as is.

At the same time, the analysis presented in response to Q10 in the rebuttal letter and Figure R2 are quite interesting and can be included as an additional SI figure and referenced to support the statement about the sensing region of the channel being located in the first constriction.

Reviewer #2:

Remarks to the Author:

All my comments have been addressed. Publish as is.